# Multiple F-Box Proteins Collectively Regulate Cell Development and Pathogenesis in the Human Pathogen *Cryptococcus neoformans*

**DOI:** 10.3390/jof8121259

**Published:** 2022-11-29

**Authors:** Chengjun Cao, Yina Wang, Samantha L. Avina, John Walter, Chaoyang Xue

**Affiliations:** 1Public Health Research Institute, New Jersey Medical School, Rutgers University, Newark, NJ 07103, USA; 2Department of Microbiology, Biochemistry and Molecular Genetics, New Jersey Medical School, Rutgers University, Newark, NJ 07103, USA; 3Rutgers Center for Lipid Research, Rutgers University, New Brunswick, NJ 08901, USA

**Keywords:** *Cryptococcus neoformans*, F-box protein, E3 ligase, fungal pathogenesis, stress response

## Abstract

The ubiquitin–proteasome system (UPS) mediates intracellular proteins degradation that influences various cellular functions in eukaryotic cells. The UPS is also involved in the development and virulence of pathogenic fungi. F-box proteins, which are part of the SCF (Skp1-Cullin-F-box protein) ligase, are a key component of UPS and are essential for the recognition of specific substrates. In this study, we identified 20 F-box proteins in *C. neoformans* and obtained deletion mutants for 19 of them. A comprehensive phenotypic analysis of these mutants revealed the diverse function of F-box proteins in stress response, cell size regulation, sexual reproduction, antifungal drug resistance, and fungal virulence in *C. neoformans*. The importance of three F-box proteins: Fbp4, Fbp8, and Fbp11, in these cellular functions were characterized in detail. This study provides an overall view of the F-box gene family in *C. neoformans*, which will lead to a better understanding of the function of fungal SCF E3 ligase-mediated UPS in fungal development and pathogenesis.

## 1. Introduction

Invasive fungal infections kill ~1.5 million people each year and are a significant global public health burden [1]. Due to the limited drug options for treating fungal infections and increased drug resistance, elucidating the regulatory mechanisms of important cellular functions and developing novel drug targets to combat fungal infections are critical. *Cryptococcus neoformans* is a yeast pathogen and the most common cause of fungal meningitis in the immunocompromised population which includes individuals with HIV infection, organ transplant recipients, and patients undergoing immunosuppressive therapy [2,3]. The global burden of cryptococcal meningitis accounts for over 180,000 deaths each year, representing ~15% of AIDS-related deaths [4]. The systemic infection of *Cryptococcus* first occurs through inhalation of spores or small yeast cells from the environment into the lung. Subsequently, the yeast disseminates to the central nervous system to cause life-threatening meningoencephalitis [5]. *C. neoformans* produces several virulence factors, including polysaccharide capsule, melanin, the ability to grow at 37 °C, and cell enlargement to adapt to host conditions [6,7]. Thus, understanding how fungal cells regulate these virulence factors in the context of fungal pathogenesis is critical for discovering novel drug targets and developing specialized therapeutic approaches [8,9,10].

The ubiquitin–proteasome system (UPS) is a key regulatory mechanism for cellular processes in eukaryotes, and has been well studied in several model organisms, including the model yeast *Saccharomyces cerevisiae* [11,12]. The UPS-mediated protein turnover includes a series of enzymatic reactions, including the activation of ubiquitin by the ubiquitin-activating enzyme E1, the transfer of activated ubiquitin to the ubiquitin-conjugating enzyme E2, and then ubiquitin binding to target substrates by interacting with ubiquitin ligase E3. Repeating this process leads to polyubiquitination of the targeted proteins, which are then degraded into small peptides and reusable ubiquitin in the 26S proteasome complex [11,12]. The UPS-mediated rapid protein turnover plays multiple key roles in fungal virulence and host adaptation [13,14,15,16,17,18,19,20]. The Skp1-Cullin-F-box protein (SCF) ligase complex is the largest E3 ligase family in which the exchangeable F-box proteins directly bind to target proteins and are responsible for substrate specificity [21]. Studies in medically and agriculturally important fungal pathogens (e.g., Species of *Aspergillus*, *Candida*, *Cryptococcus*, *Fusarium* and *Magnaporthe*, etc.) have revealed that many F-box proteins, including Cdc4 and Grr1 homologs, are required for fungal virulence [13,14,19,22,23,24,25,26].

*C. neoformans* has at least 20 proteins containing the F-box domain. Our previous studies have characterized Fbp1, which contains an F-box domain and 12 leucine rich repeats [13,27,28,29]. Fbp1 is essential for fungal virulence, as the *fbp1*∆ mutant infection induced a superior Th1 protective immunity that prevents fungal dissemination to cause disease [13,28]. Interestingly, mice vaccinated with heat-killed *fbp1*Δ cells develop a robust Th1 response and acquire protection against infection with *Cryptococcus* species and *Aspergillus fumigatus* [28,29]. The function of many other F-box proteins in this fungus remains unknown.

In this study, we set out to investigate the function of other F-box proteins identified in *C. neoformans* by mutagenesis. Characterizing all 19 mutants, we found that three F-box proteins regulate virulence factor development and are required for fungal virulence. These proteins include two Cdc4 homologs (Fbp4 and Fbp11) and one novel protein Fbp8. In a recent study, Fbp4 (CNAG_00693), named Cdc4 due to its sequence homology with Cdc4 in *S. cerevisiae* and *C. albicans*, was identified as a regulator of cell membrane integrity, stress response, sexual reproduction, and fungal virulence [22]. We identified a second Cdc4 homolog, Fbp11 (CNAG_05294) and found redundant yet distinct roles of Fbp4 and Fbp11 in regulating cell size and fungal pathogenicity in *C. neoformans*. Deletion of *FBP4* or *FBP11* increased cell body size and titan cell production. While the *fbp4*∆ mutant showed enlarged capsule size, cell membrane defects, and failed to produce spores, *fbp11*∆ mutant had melanin production defect. Both the *fbp4*∆ mutant and the *fbp11*∆ mutant show attenuated virulence in a murine inhalation model of cryptococcosis. Additionally, we identified that Fbp8 (CNAG_04341) is required for cell membrane integrity and fungal virulence. Our work illustrates the overall function of F-box proteins in *C. neoformans*.

## 2. Results

### 2.1. Identification of F-Box Proteins in C. neoformans

To understand the role of F-box proteins in fungal pathogenicity in *C. neoformans*, we scanned the genome sequence of *C. neoformans* strain H99 and identified 20 proteins containing an F-box domain in the H99 genome database (https://fungidb.org/fungidb, accessed on 20 March 2020) (Table 1). Among them, only Fbp1 contains Leucine Rich Repeats (LRRs) and 7 proteins containing WD40 repeats. We numbered the 20 F-box proteins from Fbp1 to Fbp20 based on the Gene ID, with the previously published Fbp1 (CNAG_05280) used as a control. We obtained 14 mutants from a genome-wide *Cryptococcus* deletion library [30] and confirmed them using colony-PCR (Appendix A). We also generated deletion mutants for five of the remaining six genes in wild type strain KN99α (Table 1) (Fbp6-CNAG_03157, Fbp7-CNAG_03421, Fbp12-CNAG_05450, Fbp13-CNAG_05454, Fbp19-CNAG_07551) using electroporation (Appendix A) and colony-PCR screening (Appendix A). We found Fbp14 (CNAG_05773) may be an essential gene in *C. neoformans* as its homolog Met30 is essential in *S. cerevisiae* [31] and were unable to generate a deletion strain after several attempts.

### 2.2. Phenotypes of F-Box Protein Encoding Genes Deletion Mutants under In Vitro Conditions

To elucidate functions of F-box proteins in adaptation and virulence of *C. neoformans*, we examined phenotypic traits under distinct in vitro conditions: nitrosative and oxidative stress (NaNO_2_, H_2_O_2_), osmotic stress (KCl, NaCl, Sorbitol), cell membrane and cell wall integrity (SDS, CFW, Congo red), and virulence factors (melanin, capsule, and cell size) (Appendix A and Figure 1). The phenotypic data set of all 19 mutants were qualitatively illustrated by color scale, and the phenotype of KN99α was performed as a control (Figure 1). We found that 11 mutants have at least one phenotype and 8 mutants with modest phenotypic change compared to wild type strain. The *fbp8*∆ mutant (CNAG_04341) possesses an SDS sensitivity phenotype in the conditions we tested. *fbp4*∆ (CNAG_00693) and *fbp11*∆ mutants (CNAG_05294) showed multiple phenotypic changes in the virulence and stress response to environment including cell membrane integrity, cell size development, and melanin production. Fbp4 and Fbp11 are two F-box and WD-40 domain-containing proteins that share protein sequence similarity with Cdc4 in *S. cerevisiae* (35% and 32%, respectively), suggesting that Fbp4 and Fbp11 are two Cdc4 homologs in *C. neoformans*. To identify the functional correlation between Fbp4 and Fbp11, we generated a *fbp4*∆ *fbp11*∆ double mutant and investigated the function of Fbp4, Fbp11, and Fbp8 as their deletion mutants demonstrate clear phenotypes.

### 2.3. F-Box Proteins Are Important for Cell Membrane Integrity and Virulence Factor Production

To assay for cell membrane integrity defects, we treated mutants with 0.03% SDS as previously described [13]. We found that both the *fbp4*∆ and *fbp4*∆ *fbp11*∆ mutants were hypersensitive to SDS, whereas the *fbp11*∆ single mutant showed no sensitivity difference compared to the wild type strain. These observations suggest a specific role of Fbp4 in cell membrane integrity (Figure 2A). Additionally, the *fbp8*∆ mutant showed increased sensitivity to SDS compared to wild type (Figure 2A). To demonstrate that this loss of function in cell membrane integrity was due to F-box protein deletion, we tested complemented strains of these mutants (*fbp4*∆ + *FBP4*, *fbp11*∆ + *FBP11*, and *fbp8*∆ + *FBP8*). Indeed, complemented strains of the F-box protein deletion mutants restored cell membrane integrity indicated by normal growth in SDS condition (Figure 2A).

Several virulence factors, such as the polysaccharide capsule and the polyphenol pigment melanin, are essential for *C. neoformans* virulence [6,7,37,38]. We examined the melanin production of these F-box protein mutants on both L-DOPA (L-3,4-dihydroxyphenylalanine) and Niger seed agar media. The *fbp11*∆ and *fbp4*∆ *fbp11*∆ mutants exhibited a melanin production defect (Figure 2A and Appendix A), whereas the *fbp4*∆ and *fbp8*∆ mutants produced normal melanin on both media (Figure 2A and Appendix A). We investigated capsule development of F-box protein mutants on DME agar medium at 30 °C (Figure 2B and Appendix A). Quantitative measurement of capsule thickness (Figure 2C) and relative capsule size (Figure 2D) showed that deletion of either *FBP4* or *FBP11* increased capsule production and the *fbp4*∆ + *FBP4* or *fbp11*∆ + *FBP11* reconstituted strain reverted to normal capsule size. These results indicate an important role of Fbp4 in capsule size regulation. Additionally, we observed the capsule enlargement of the *fbp9*∆ mutant on DME agar medium at 37 °C (Appendix A).

### 2.4. Fbp4 and Fbp11 Negatively Regulate Cell Size

Titan cell formation has been identified as an important virulence factor in *C. neoformans* [39,40,41]. Our previous study demonstrated that Fbp1 negatively regulates yeast cell size and the *fbp1*∆ strain significantly induces titan cells compared to wild type in both in vitro conditions and in vivo from infected lungs [42]. Similarly, the *fbp4*∆ mutant cells were significantly larger than the wild type cells when grown on capsule-inducible medium DME (Figure 2B, Figure 3A and Appendix A). The *fbp4*∆ *fbp11*∆ double mutant also showed increased cell size similar to the *fbp4*∆ single mutant (Figure 3A). We then measured the cell body size and calculated the large cell percentage of these mutants under in vitro titan cell inducing conditions as previously reported [43] (Figure 3C,D). Here, *fbp4*∆ had a larger cell body size (8.1 µm in *fbp4*∆ vs. 6.6 µm in WT, *p* = 0.003) and a higher percentage of titan cells formation than wild type (34.1% in *fbp4*∆ vs. 18.0% in WT, *p* < 0.01). Additionally, *fbp11*Δ mutant increased the median cell body size and titan cell production (Figure 3C,D). This phenotype was fully rescued in the reconstituted compliment strains. Other mutants, including *fbp8*∆, had no significant cell size changes. Interestingly, the *fbp4*Δ *fbp11*Δ double mutant produced more titan cells compared to the *fbp4*Δ or the *fbp11*Δ single mutant (Figure 3D). These results indicate that Fbp4 and Fbp11 have overlapping yet additive roles in titan cell production.

### 2.5. Fbp8 and Fbp4 Are Important for Fluconazole Resistance

To illustrate the potential role of F-box proteins on regulation of antifungal sensitivity, we performed an in vitro susceptibility assay for all the F-box protein mutants to measure the MIC_90_ of three antifungal drugs: fluconazole, caspofungin, and amphotericin B. MIC_90_ testing showed that both the *fbp8*Δ and *fbp4*Δ mutants are both highly susceptible to fluconazole with a MIC_90_ at 2 µg/mL, compared to the wild type MIC_90_ of 8 µg/mL (Table 2). The *fbp4*Δ *fbp11*Δ double mutant had the same fluconazole sensitivity as the *fbp4*Δ mutant. Meanwhile, all mutants had same MIC_90_ value for caspofungin and amphotericin B as wild type (Table 2). To test the hypothesis that Fbp4 and Fbp8 regulates the ergosterol biosynthesis as a mean of fungal resistance to triazole drugs, we performed qRT-PCR to measure the expression of *ERG11* in wild type and the *fbp4∆* and *fbp8∆* mutants. Interestingly, *fbp4*Δ and *fbp8*Δ did not show a significant difference in *ERG11* expression level compared to wild type when grown at half MIC_90_ fluconazole conditions. These findings suggest the regulation of ergosterol biosynthesis may not be at the transcription level or involves other mechanisms which target fluconazole susceptibility in these mutants (Appendix A).

### 2.6. Fbp4 Is Essential for Sexual Reproduction

To evaluate the function of F-box proteins in sexual reproduction, we performed genetic cross between KN99**a** and mutants of F-box genes in KN99α background to generate mutants in opposite mating type and obtained all F-box protein mutants in both α and **a** mating types. We examined the development of dikaryotic hyphae and basidiospores in both unilateral crosses and bilateral crosses for all mutants of the F-box genes (Appendix A). A mating cross between KN99α and KN99**a** was performed as the control (Appendix A). We did not observe a clear defect in mating hyphae production or sporulation in unilateral crosses of all mutants. The bilateral mating between *fbp4*Δ mutants produced normal mating hyphae but failed to produce basidiospores, which is consistent with a previous finding [22].These data suggest that Fbp4 and Fbp11 play distinct roles in sexual reproduction (Appendix A).

### 2.7. F-Box Proteins Are Involved in Cryptococcus-Macrophage Interaction

As a facultative pathogen, *C. neoformans* can survive and proliferate inside macrophages, which is important for *Cryptococcus* dissemination [44,45,46]. We investigated the potential role of F-box proteins in *Cryptococcus*-macrophage interaction with macrophage cell line RAW264.7 following a previously described protocol [27] (Figure 4 and Appendix A). Non-adherent extracellular yeast cells were removed after 2 h of *Cryptococcus*-macrophage coincubation. The phagocytosis assay and macrophage killing assay were performed after incubation for an additional 0, 2, and 24 h. We found that Fbp8 was not required for *Cryptococcus*-macrophage interaction (Figure 4A), while deletion of *FBP4* or *FBP11* significantly reduced intracellular growth in macrophages after 24 h coincubation compared to the wild type strain. The complement strains completely restored the survival in macrophages (Figure 4B,C). The CFUs (Colony forming units) recovered from the *fbp4*Δ *fbp11*Δ double mutant engulfed by macrophages were significantly less than either the *fbp4*Δ or *fbp11*Δ single mutants (Figure 4D). These results demonstrate that Fbp4 and Fbp11 are important for fungal survival in macrophages. In addition, we observed that the *fbp7*Δ mutant had proliferation defect inside macrophages while deletion of *FBP9* induced *C. neoformans* proliferation in macrophages, both are interesting observations that warrant further investigation (Appendix A).

### 2.8. Roles of F-Box Proteins in Fungal Virulence

Because of the importance of Fbp4, Fbp11, and Fbp8 in development of *Cryptococcus* virulence factors, we assessed the virulence of the *fbp4*Δ, *fbp11∆*, *fbp8*Δ, and *fbp4*Δ *fbp11*Δ mutants in a murine inhalation model of systemic cryptococcosis. Mice were infected with 10^5^ cells of each strain and surveilled for survival following infection. Mice infected with *fbp8*Δ and *fbp11*Δ mutants succumbed to infection within 32 and 38 days post-inoculation, respectively, while those infected with wild type strain died within 25 days as expected (Figure 5). These results indicated that *fbp8*Δ and *fbp11*Δ have significant virulence attenuation. The *fbp4*Δ-infected mice survival to 54 dpi (days post infection) also indicated significant virulence attenuation (Figure 5B). Excitingly, mice infected with the *fbp4*Δ *fbp11*Δ double mutant survived to the termination point of the experiment (60 dpi) with no visible symptoms (Figure 5B), suggesting Fbp4 and Fbp11 share an additive function in regulating fungal virulence.

## 3. Discussion

F-box proteins determine the substrate specificity of the UPS and play multiple important roles in fungal development and virulence [14,47]. *C. neoformans* possesses at least 20 F-box proteins including Fbp1 [13,28,29]. In this study, we obtained mutants of 19 F-box protein encoding genes, including 14 mutants from the *Cryptococcus* deletion library generated by Dr. Madhani group [30] and five new mutants generated in this study. We could not generate the mutant for *FBP14* (homolog of *MET30*) and suspected it to be an essential gene in *C. neoformans*. Fbp1 has been studied extensively [13,27,28,29,42], and thus used as a control in this study. We studied functions of the remaining 18 non-essential F-box proteins by analyzing their respective mutants under a range of stress conditions. Here, we identified three F-box proteins (Fbp4, Fbp8, and Fbp11) with important cellular functions based on clear phenotypes of their mutants, including sexual reproduction, cell size regulation, drug resistance, and fungal virulence. Fbp4 and Fbp11 share high protein structure and sequence similarity with Cdc4 in *S. cerevisiae*, suggesting that both are Cdc4 homologs in *C. neoformans*. We also found Fbp4 and Fbp11 share overlapping and distinct cellular functions, including cell integrity, stress response, cell size regulation, capsule and melanin production, and fungal–host interactions. Fbp8 is required for fungal cell membrane integrity as its mutant is hypersensitive to SDS treatment, but not cell wall stressing agents calcofluor white and congo red.

Cryptococcal adaptation to the hostile host environment is critical for its virulence. The *fbp4*Δ mutant exhibited increased susceptibility to oxidative stress H_2_O_2_, as well as cell membrane stressor SDS, which may interfere with the ability of the fungal proliferation inside macrophages. The UPS regulation of oxidative stress responses has been identified across multiple eukaryotic species [48,49]. For example, deletion of *UBI4* shows hypersensitivity to H_2_O_2_ in *S. cerevisiae* [50]. Two ubiquitin-conjugating enzymes Ubc6-2 and Ubc8 and a deubiquitinating enzyme Ubp5 are involved in diverse stress responses in *C. neoformans* [49,51]. We found that both *fbp4*Δ and *fbp8*Δ mutants are sensitive to SDS, but not the cell wall stressors such as calcofluor white and congo red. These findings indicate that Fbp4 (Cdc4 in a separated report [22]) and Fbp8 are involved in cell membrane integrity similar to Fbp1 [13,22]. It is possible that some substrates of these F-box proteins are important for membrane integrity. Indeed, homolog of Fbp8 in *S. cerevisiae* (Rcy1) is involved in recycling plasma membrane proteins. Rcy1 directly interacts with C-terminal region of phospholipid flippase Drs2 and plays an important role in the function of Drs2 in *S. cerevisiae* [52].

Consistent with the SDS sensitivity phenotype of the *fbp4*Δ and *fbp8*Δ mutants, we also observed that both Fbp4 and Fbp8 are required for fungal resistance to azole drug fluconazole, but not for other classes of antifungal agents including caspofungin and amphotericin B. This suggested that Fbp4 and Fbp8 may be involved in regulation of the ergosterol biosynthesis pathway. As overexpression of *ERG11* gene leads to overproduction of the target enzyme of fluconazole and increase drug resistance [53], we tested the role of Fbp4 and Fbp8 in the regulation of *ERG11* expression. However, we observed no expression change of *ERG11* in the *fbp4*Δ and *fbp8*Δ mutants compared to their parental wild type and complement strains. It is possible that these F-box proteins regulate Erg11 through post-translational modifications (e.g., ubiquitin-mediated proteolysis) rather than transcriptional regulation. In addition, alterations in efflux pumps may play a role in azole susceptibility in the forementioned F-box protein mutants [54]. Alteration of drug accessibility to the target enzyme by regulation of efflux pumps (including CDR and MDR genes) has been shown to be an important mechanism of azole resistance in *Candida* species [53,54]. Although previous studies have indicated that efflux pumps (Afr1 and Afr2) only play a minor role in *C. neoformans* resistance to azoles [55,56], it remains possible that they contribute to F-box protein-mediated azole resistance. Overall, the mechanism of increased azole sensitivity in these F-box protein mutants remains to be further investigated. In *S. cerevisiae*, Rcy1 is required for the Drs2 function in the early endosome to the trans-Golgi network (TGN) transport [52]. Cdc50-Drs2 function as lipid translocase (flippase) generates membrane phospholipid asymmetry and regulates ergosterol distribution [57,58,59]. In *Cryptococcus*, Cdc50 is the only homolog of the ScCdc50 gene family and the *cdc50*Δ mutant sensitive to fluconazole [60]. As there is a similar sensitivity with the *fbp8*Δ mutant, these findings suggest Fbp8 may also regulate lipid flippase function in *C. neoformans*.

Among the three well studied virulence factors in *C. neoformans,* none of these mutants exhibit significant growth defect at a host body temperature of 37 °C. A melanin defect in the *fbp11*∆ mutant, but not in the *fbp4*∆ mutant, suggests that although both Fbp4 and Fbp11 are homologs of Cdc4 in *S. cerevisiae*, they have distinct functions. Yet, it is interesting that both the *fbp4*∆ mutant and the *fbp11*∆ mutant showed increased capsule sizes. Further study is warranted to understand how these two F-box proteins regulate capsule growth, which is beyond the scope of this study.

Titan cell production (cell body size over 10 µm) can be observed during cryptococcal infection and has been identified as a virulence factor [39,40,61]. The Gpa1-cAMP/PKA signal pathway is critical in regulating cell size change during *Cryptococcus* infection [41,62]. We found that both Fbp4 and Fbp11 are involved in cell size regulation, and that the deletion of *FBP4* or *FBP11* induces titan cell production. The *fbp4*Δ *fbp11*Δ double mutant exhibits greater titan cell percentage in a population than either single mutant. In *S. cerevisiae*, Cdc4 is required for the G1/S and G2/M phase transitions [63,64]. Because dysregulation of the cell cycle has been shown to be involved in titan cell production in *C. neoformans* [65], it is not surprising that Fbp4 and Fbp11 are involved in cell size regulation. It has been reported that the meiosis machinery is involved in ploidy reduction during titan cell proliferation [66]. Our previous work found that the CDK-dependent protein kinase Crk1, a meiosis regulator, positively regulates titan cell production in *C. neoformans* [42]. Deletion of Fbp4 blocks sporulation but has no impact on other events in bilateral mating, such as cell fusion, dikaryotic hyphae formation, and nuclear fusion in *C. neoformans* [67]. These findings similar to the *fbp1*Δ mutant [13] and the published data on Cdc4 [22] thus indicate Fbp4 may play a key role in meiosis regulation. Since both *Saccharomyces* Cdc4 and *Cryptococcus* Fbp4 are involved in sexual reproduction [68,69], it is possible that Fbp4 may also regulate titan cell production through the meiosis machinery.

Despite producing larger capsule sizes in the *fbp4*Δ and *fbp11*Δ mutants, both mutants and the *fbp4*Δ *fbp11*Δ double mutant showed virulence attenuation in a murine inhalation model of cryptococcosis. Together with our previous findings on Fbp1 function, our studies further demonstrate the importance of F-box proteins in pathogenicity of *C. neoformans*. We found that fungal virulence of these three mutants were consistent with their ability of intracellular proliferation in macrophages. Wu et al. reported that Cdc4 (Fbp4) was involved in the virulence in *C. neoformans* through similar mechanisms as Fbp1 and that the loss of *CDC4* blocked *C. neoformans* dissemination from lung to other organs and infected mice survived for over two months [22]. In our study, albeit virulence attenuation, *fbp4*Δ-infected mice still succumb to lethal infection between 30 and 54 dpi. The difference may be due to different *Cryptococcus* strain and mouse strain backgrounds used in these studies. Specifically, our *fbp4*Δ mutant was generated from the KN99α background strain and tested for virulence in the BALB/c mouse strain, while Wu and colleagues utilized the C57BL/6 mouse strain and *C. neoformans* H99 background strain in their studies. The *fbp4*∆ *fbp11*∆ double mutant showed significantly more severe virulence attenuation than either of the single mutants, indicating these two Cdc4 homologs are both required for fungal virulence and have additive functions. In aggregate, our systemic characterization of the entire F-box gene family in *C. neoformans* reveals a diverse role of F-box proteins in *Cryptococcus* development and virulence. Further study on the molecular detail of SCF E3 ligase-mediated proteolysis may lead to novel drug targets in fungal pathogens.

## 4. Materials and Methods

Strains and media: *Cryptococcus neoformans* gene deletion libraries were generated by Hiten Madhani at UCSF and purchased from the Fungal Genetic Stock Center (FGSC). Other strains used in this study are listed in Appendix A. YPD medium (2% peptone, 1% yeast extract, 2% glucose) was used for routine culture of *C. neoformans* strains. Stress media were created by adding different stress-inducing agents into YPD agar medium. L-DOPA (L-3,4-dihydroxyphenylalanine) and Niger seed media were used to test melanin production. Dulbecco’s modified Eagle’s (DME) medium for assessing capsule production was prepared as previously described [70]. V8 juice agar medium (50 mL V8 original juice, 0.05% KH_2_PO_4_, 2% agar, pH 5.0) and modified Murashige and Skoog (MS) medium minus sucrose (Sigma-Aldrich, Steinhelm, Germany) were used for mating and sporulation assays [71]. Minimal medium (MM, 15 mM D-glucose, 10 mM MgSO_4_, 29.4 mM KH_2_PO_4_, 13 mM Glycine, 3.0 μM Thiamine) was used for titan cell induction [43]. Macrophage murine cell line RAW264.7 was grown in liquid Dulbecco Modified Eagles (DME) medium with 10% heat inactivated fetal calf serum (FBS) (ATL Biologicals) and prepared as previously described [71].

Electroporation: Electroporation was performed as previously described [72]. Briefly, overnight cultures were transferred into 100 mL YPD at an initial OD_600_ around 0.2 and grown for 4–5 h at 30 °C until the OD_600_ reach 0.6–0.8. All samples were kept cold in the following steps. Cells were collected, washed twice, resuspended in electroporation buffer (10 mM Tris-HCl pH 7.5, 1 mM MgCl_2_, 270 mM sucrose), and incubated with 1 mM DTT on ice for one hour. Cells were pelleted and resuspended in 250 μL electroporation buffer. 45 μL cells were mixed with 5 μL DNA and transferred to a precooled 2 mm gap cuvette. An Eppendorf multiporator was used for electroporation with the following settings: V = 2 kv. Cells were resuspended in 1 mL YPD, transferred to 1.5 mL Eppendorf tubes, and incubated at 30 °C for 1.5 h before plating onto a selective agar plate.

Colony PCR: Colony PCR was conducted on *Cryptococcus* cultures grown on YPD agar plate. Small colonies were picked and transferred to PCR tubes as direct PCR template. Colony PCR bands were checked by gel electrophoresis in 1% agarose gel.

RNA extraction and qRT-PCR: Total RNA was extracted from *C. neoformans* strains with Trizol (Invitrogen). Extracted RNA was treated with rDNAse kit and purified using the Nucleospin RNA cleanup kit (Macherey–Nagel). Relative mRNA levels were determined by quantitative reverse transcription (qRT) PCR. *Cryptococcus* strains were grown at a 1.0 OD for 4 h at 30 °C with 1μg/mL of fluconazole. An amount of 2 μg of purified RNA were converted to cDNA using SMARTScribe Reverse Transcriptase cDNA synthesis kit (Takara Bio). The SYBR Green PCR Master mix (Applied Biosystems) and constructed primers for each gene were used and normalized to the gene encoding glyceraldehyde-3-phosphate dehydrogenase (GAPDH). Gene expression relative to parental KN99α grown in the same conditions was calculated by the ΔΔCT method.

Phenotypic assays. (i) Cell wall and membrane integrity: Overnight cultures with serial dilutions were placed on YPD agar plates containing 0.03% SDS to test cell membrane integrity. YPD with the supplement of 0.5% congo red, or 250 μg/mL calcofluor white (CFW) were used to test cell wall integrity. The agar plates were incubated at 30 or 37 °C for 3 days before being photographed.

(ii) Melanin and capsule production: Overnight cultures with serial dilutions were placed on (L-DOPA) agar medium or overnight cultures were inoculated on Niger seed agar medium to measure the melanin production. The agar plates were incubated at 30 or 37 °C for 3 days before being photographed. Overnight cultures were inoculated on Dulbecco modified Eagle (DME) medium and incubated at 37 °C for 7 days to examine capsule production. The capsule size was visualized by adding a drop of India ink to the cell suspensions and observed on an Olympus AX70 microscope (Melville, NY, USA). The relative capsule size was calculated by dividing capsule size with the whole-cell size (capsule and cell size). The average and standard deviation from at least 50 cells were calculated for each condition tested.

(iii) Assays of stress responses: Yeast cells from overnight cultures were washed, resuspended, and serially diluted (1:10) in ddH_2_O and spotted (5 µL) on YPD agar plates containing 1.0 M KCl, 1.5 M NaCl, or 2.5 M sorbitol for osmotic shock, 5 mM H_2_O_2_ for oxidative stress, and 1 mM NaNO_2_ for nitrosative stress as previously described [60].

(iv) In vitro titan cell induction and measurement: Titan cell inductions were performed as reported previously [43]. Strains were grown overnight at 30 °C in YPD liquid medium. Cells were collected, washed twice with minimal medium, and suspended in minimal medium with the final concentration of 10^6^ cells/mL. A 1 mL suspension was incubated at 30 °C for 3 days in a 1.5 mL Eppendorf tube with the cap closed. Body sizes of more than 100 cells were measured and the titan cell (cell body size over 10 µm) proportions were calculated.

(v) Assays of mating: *C. neoformans* cells of opposite mating types were mixed and cultured on V8 or MS agar medium at 25 °C in the dark. Mating filaments and basidiospores formation were examined and recorded by photography after incubation for 7 or 14 days.

*Cryptococcus*-macrophage interaction assay: Macrophage cell line RAW264.7 cells were cultured in a DME medium with 10% heat inactivated FBS at 37 °C with 5% CO_2_. A total of 5 × 10^4^ macrophage cells in 0.5 mL fresh DME was put into each well of a 48-well culture plate and incubated at 37 °C in 5% CO_2_ overnight. To activate macrophage cells, 50 unit/mL gamma interferon (IFNγ, Invitrogen) and 1 μg/mL lipopolysaccharide (LPS, Sigma) were added into each well. *C. neoformans* overnight cultures were washed with PBS twice and opsonized with 20% mouse complement. A total of 2 × 10^5^
*Cryptococcus* cells were added into each well (yeast/macrophage ratio, 4:1). To assess intracellular proliferation of *C. neoformans*, non-adherent extracellular yeast cells were removed by washing with fresh DME medium after 2 h co-incubation and cultures were incubated for another 0, 2, and 24 h. At indicated time points, the medium in each well was replaced by ddH_2_O to lyse the macrophage cells for 30 min at room temperature. The lysate was spread on YPD plates and CFU was counted to determine intracellular proliferation.

Virulence studies: Yeast strains were grown at 30 °C overnight and washed twice with PBS buffer and resuspended to a final concentration of 2 × 10^6^ cells/mL. Groups of 10 female BALB/c mice were intranasally infected with 10^5^ yeast cells of each strain as previously described [73]. Over the course of the experiments, animals that appeared moribund or in pain were sacrificed by CO_2_ inhalation. Survival data from the murine experiments were statistically analyzed between paired groups using the long-rank test using the PRISM program 8.0 (GraphPad Software) (*p* values of ≤0.05 were considered significant). All animal studies were conducted following biosafety level 2 (BSL-2) protocols and procedures approved by the Institutional Animal Care and Use Committee (IACUC) and Institutional Biosafety Committee of Rutgers University, respectively. The studies were conducted in facilities accredited by the Association for Assessment and Accreditation of Laboratory Animal Care (AAALAC).

## Figures and Tables

**Figure 1 jof-08-01259-f001:**
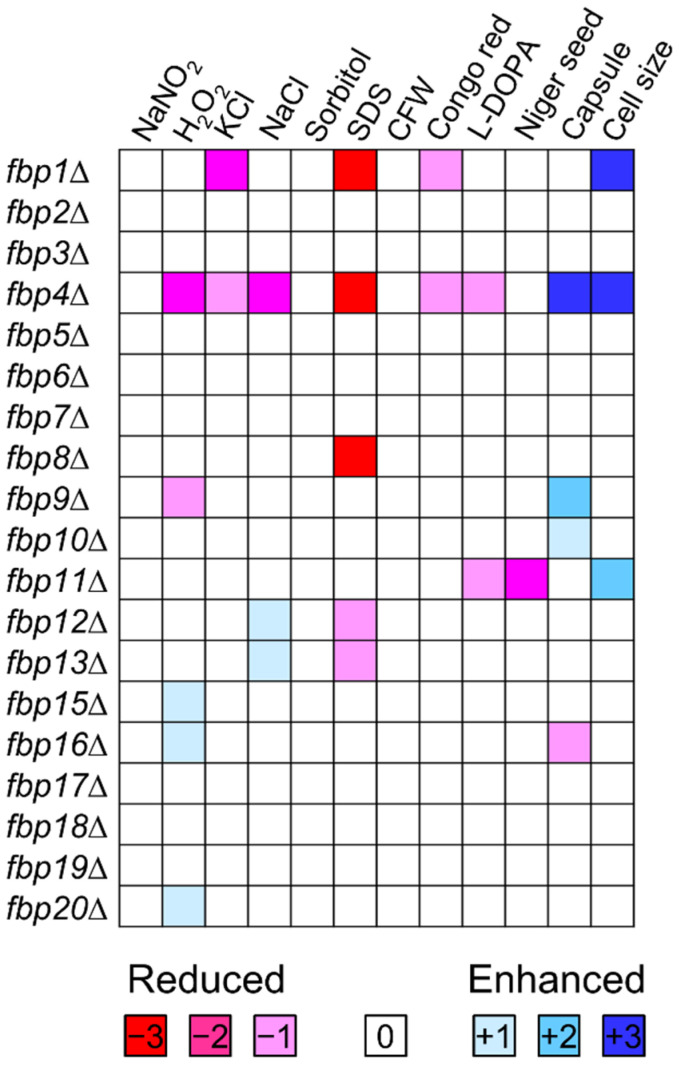
Phenotypic clustering of F-box proteins in *C. neoformans*. Phenotypic assays were examined under different conditions and scored on a 7-point scale. −, reduced. +, enhanced.

**Figure 2 jof-08-01259-f002:**
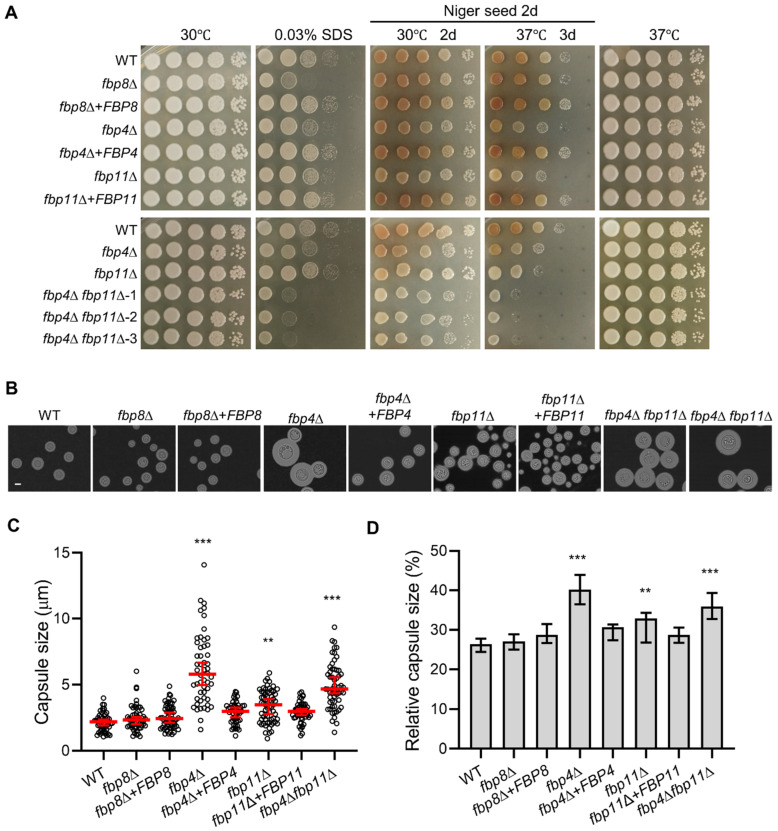
F-box proteins are required for cell integrity and development of virulence factors. (**A**) Cell membrane stress and melanin production assays. Overnight cultures were collected and resuspended to an optical density at 600 nm (OD600) of 10.0. Tenfold serial dilutions were made in ddH_2_O, and 5 μL of each were plated on 0.03% SDS or Niger seed agar. Plates were incubated at 37 °C or 30 °C for 2 or 3 days. (**B**) Representative images to show capsule formation at 37 °C on DME medium. Capsule production was visualized by India ink staining after cells were grown on DME medium for 7 days. Bar, 5 µm. (**C**) Capsule size was measured by India ink staining. (**D**) Relative capsule size was calculated by dividing observed capsule size by whole-cell size. The average and standard deviation from at least 50 cells were calculated for each condition tested. WT, wild type. Statistical analysis for all measurements in this figure was performed with the Kruskal–Wallis nonparametric test for multiple comparisons. **, *p* ≤ 0.01. ***, *p* ≤ 0.001.

**Figure 3 jof-08-01259-f003:**
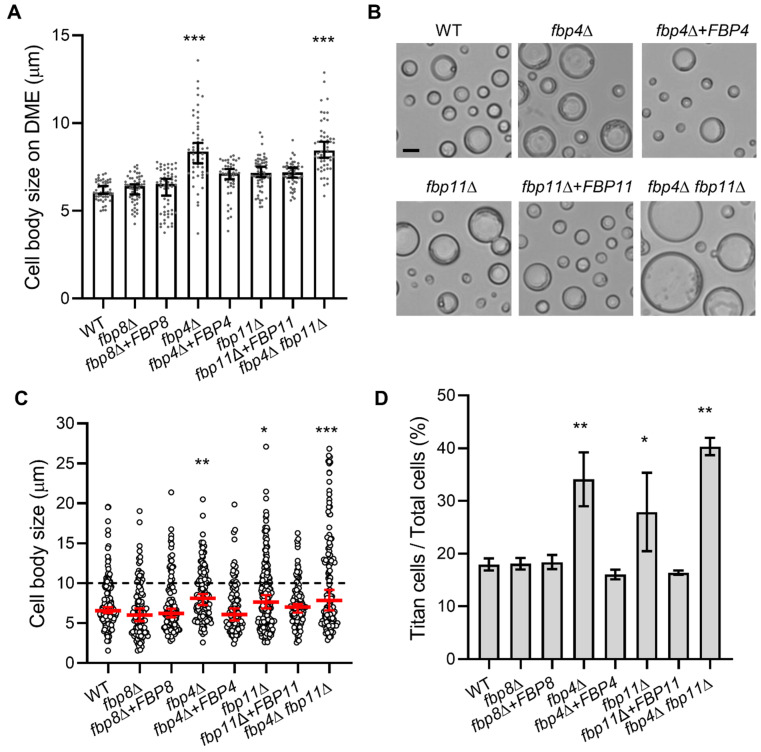
Fbp4 and Fbp11 are required for cell size control. (**A**) Quantitative measurement of cell body size grown on DME medium for 7 days. Error bar indicates the standard deviation of the mean for more than 50 cells. (**B**) Representative images of WT, *fbp4*Δ, *fbp4*Δ + *FBP4*, *fbp11*Δ, *fbp11*Δ + *FBP11*, and *fbp4*Δ *fbp11*Δ strains under in vitro titan cell inducible condition. Bar, 10 µm. (**C**,**D**) Quantitative measurement of cell size (**C**) and titan cell percentage (**D**) of strains cultured under in vitro titan cell inducing conditions. Error bar indicates the standard deviation of the mean for more than 100 cells. Statistical analysis for all measurements in this figure was performed with the Kruskal–Wallis nonparametric test for multiple comparisons. *, *p* ≤ 0.05. **, *p* ≤ 0.01. ***, *p* ≤ 0.001.

**Figure 4 jof-08-01259-f004:**
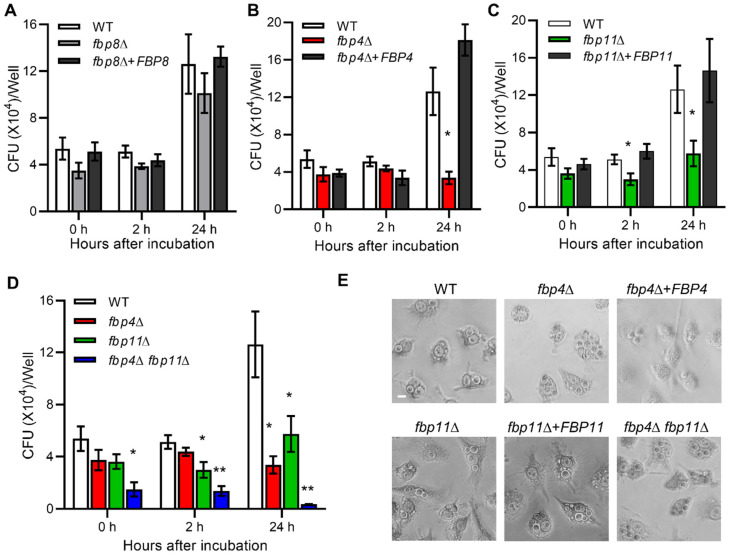
Fbp4 and Fbp11 are important for *Cryptococcus*-macrophage interactions. Statistical analysis for all measurements in this figure was performed with the Kruskal–Wallis nonparametric test for multiple comparisons. *, *p* ≤ 0.05. **, *p* ≤ 0.01. (**A**–**C**) Killing assay of *fbp8*Δ (**A**), *fbp4*Δ (**B**), *fbp11*Δ (**C**). Phagocytosis was allowed to occur for 2 h to measure both fungicidal and fungistatic activity. After each time interval, macrophages were lysed by adding sterile ddH_2_O. Cells were plated on YPD agar plates and counted to determine CFU. (**D**) Killing assay of *fbp4*Δ, *fbp11*Δ, and *fbp4*Δ *fbp11*Δ strains. (**E**) Typical field views of phagocytosis for each sample.

**Figure 5 jof-08-01259-f005:**
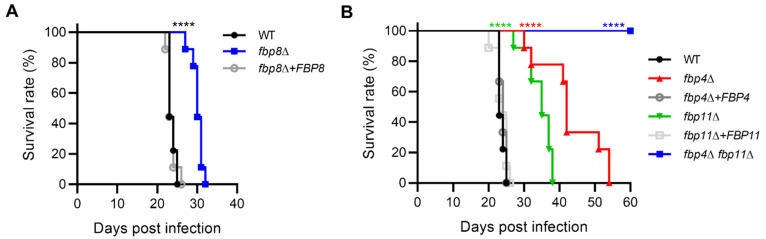
F-box proteins regulate virulence in a murine inhalation model. Survival curves for mice after intranasal infection with *fbp8*Δ (**A**), or *fbp4*Δ, *fbp11*Δ, or the *fbp4*Δ *fbp11*Δ double mutant (**B**). 6–8-week-old Female BALB/C mice (*n* = 10/group) were inoculated intranasally with 10^5^ cells of each strain and monitored for survival. Statistical analysis was performed based on Log-rank (Mantel–Cox) test. ****, *p* ≤ 0.0001.

**Table 1 jof-08-01259-t001:** The F-box proteins in *Cryptococcus neoformans*.

Name	Gene ID	Homolog in *S. cerevisiae*	Deletion Location	Function	References
Fbp1	CNAG_05280	Grr1	15C11	F-box and leucine-rich repeat protein GRR1	[13,32]
Fbp2	CNAG_00134	-	44B9	hypothetical protein	
Fbp3	CNAG_00416	-	21C12	F-box/WD-repeat protein lin-23	
Fbp4	CNAG_00693	Cdc4	34G11	F-box and WD-40 domain-containing protein CDC4	[22,33]
Fbp5	CNAG_02349	-	16H8	hypothetical protein	
Fbp6	CNAG_03157	-	KO	hypothetical protein	
Fbp7	CNAG_03421	-	KO	hypothetical protein	
Fbp8	CNAG_04341	Rcy1	31E5	recyclin-1	[34]
Fbp9	CNAG_04462	Hrt3	31F6	F-box protein 9	[35]
Fbp10	CNAG_04606	-	31H11	hypothetical protein	
Fbp11	CNAG_05294	Cdc4	33B3	F-box and WD-40 domain-containing protein CDC4	[22,33]
Fbp12	CNAG_05450	-	KO	hypothetical protein	
Fbp13	CNAG_05454	-	KO	hypothetical protein	
Fbp14	CNAG_05773	Met30	-	F-box and WD-40 domain-containing protein MET30	[31]
Fbp15	CNAG_05874	-	37A5	hypothetical protein	
Fbp16	CNAG_06382	-	38C8	beta-transducin repeat containing protein	
Fbp17	CNAG_06722	-	43F9	hypothetical protein	
Fbp18	CNAG_07482	Saf1	40C3	SCF-associated factor 1	[36]
Fbp19	CNAG_07551	-	KO	hypothetical protein	
Fbp20	CNAG_07702	-	11F3	hypothetical protein	

The gene functions and domains were searched in fungiDB (https://fungidb.org/fungidb, accessed on 20 March 2020). Abbreviation: KO: Gene knockout. Mutants can be found in *Cryptococcus* deletion collection generated by Madhani laboratory [30].

**Table 2 jof-08-01259-t002:** In vitro susceptibility of strains to antifungal drugs.

Strains	MIC_90_ (µg/mL)		
	Fluconazole	Caspofungin	Amphotericin B
WT	8	16	1
*fbp8*∆	2	16	1
*fbp8*∆ + *FBP8*	8	16	1
*fbp4*∆	2	16	1
*fbp4*∆ + *FBP4*	8	16	1
*fbp11*∆	8	16	1
*fbp11*∆ + *FBP11*	8	16	1
*fbp4*∆ *fbp11*∆	2	16	1

## Data Availability

Not applicable.

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
