# Peer review of "Multiple F-Box Proteins Collectively Regulate Cell Development and Pathogenesis in the Human Pathogen Cryptococcus neoformans"

_jof, 2022, doi:10.3390/jof8121259_

Round 1
Reviewer 1 Report
The authors build upon their prior work with the Cryptococcus neoformans Fbp1 protein to offer a comprehensive assessment of virulence-associated phenotypes for 19 of the 20 C. neoformans proteins predicted to act as F-box domain-containing SCF E3 ubiquitin ligases. The main strength of the paper is the creation of a validated group of all non-essential SCF ligases in this species, as well as detailed assessments of phenotypes commonly associated with virulence in mammals. The genetics and cell biology aspects of these experiments are exceptional, with appropriate controls and reconstituted strains when needed. The virulence experiments are equally rigorous. This manuscript will clearly serve as the basis for future experiments in protein ubiquitination and proteasomal degradation as means of regulation in this pathogenic fungus.
There is no underlying mechanistic detail offered for any of the Fbp proteins to define targets that might explain some of the phenotypes observed in the respective mutant strains. They attempt to explain the alteration in fluconazole susceptibility in selected mutants by assessing transcript levels of the ERG11 gene. While a reasonable first experiment, there are numerous reasons for altered fluconazole susceptibility, including altered activity of MDR pumps, or alterations of the stability of any component of the ergosterol biosynthesis pathway. While this would require a very detailed additional series of experiments, there should be a more informed discussion of other possible causes of this important phenotype.
There are numerous syntax errors that should be corrected by careful proof-reading. These do not detract from the overall readability, but these errors need to be carefully corrected.
Minor issues
1) Line 103-104. NaNo2 induces nitrosative, not oxidative, stress on an acidic medium. This salt is not toxic on buffered media at more neutral pH. Also consider changing the associated wording to state “oxidative/nitrosative stress sensitivity …”; and “osmotic stress sensitivity” …
2) Line 126. Consider: “These observations suggest a specific role …”
3) Line 143. I am not sure what they mean by a “redundant role” in capsule production. They use more precise and accurate wording elsewhere in the manuscript.
Author Response
The authors build upon their prior work with the Cryptococcus neoformans Fbp1 protein to offer a comprehensive assessment of virulence-associated phenotypes for 19 of the 20 C. neoformans proteins predicted to act as F-box domain-containing SCF E3 ubiquitin ligases. The main strength of the paper is the creation of a validated group of all non-essential SCF ligases in this species, as well as detailed assessments of phenotypes commonly associated with virulence in mammals. The genetics and cell biology aspects of these experiments are exceptional, with appropriate controls and reconstituted strains when needed. The virulence experiments are equally rigorous. This manuscript will clearly serve as the basis for future experiments in protein ubiquitination and proteasomal degradation as means of regulation in this pathogenic fungus.
A: We thank the reviewer for the positive response to this manuscript.
There is no underlying mechanistic detail offered for any of the Fbp proteins to define targets that might explain some of the phenotypes observed in the respective mutant strains. They attempt to explain the alteration in fluconazole susceptibility in selected mutants by assessing transcript levels of the ERG11 gene. While a reasonable first experiment, there are numerous reasons for altered fluconazole susceptibility, including altered activity of MDR pumps, or alterations of the stability of any component of the ergosterol biosynthesis pathway. While this would require a very detailed additional series of experiments, there should be a more informed discussion of other possible causes of this important phenotype.
A: Thank you for this suggestion. We added a few sentences on the possible mechanism of fluconazole susceptibility in the discussion (references 53 - 56). We will continue working on the underlying mechanisms in the future study.
There are numerous syntax errors that should be corrected by careful proof-reading. These do not detract from the overall readability, but these errors need to be carefully corrected.
A: The revised version has been carefully read to correct grammatic errors.
Minor issues
1) Line 103-104. NaNo2 induces nitrosative, not oxidative, stress on an acidic medium. This salt is not toxic on buffered media at more neutral pH. Also consider changing the associated wording to state “oxidative/nitrosative stress sensitivity …”; and “osmotic stress sensitivity” …
A: Thank you for pointing out. We corrected it.
2) Line 126. Consider: “These observations suggest a specific role …”
A: Thank you for these suggestions. We changed “indicated” to “suggest” in this sentence.
3) Line 143. I am not sure what they mean by a “redundant role” in capsule production. They use more precise and accurate wording elsewhere in the manuscript.
A: We apologize for not making it clear. We changed this sentence to: These results indicate an important role of Fbp4 in capsule size regulation.
Reviewer 2 Report
The present scientific article is original and presents good quality in the structuring of the results. It has scientific merit and is of great importance to the literature. I strongly recommend the publication of the work after adapting it to the journal's norms.
Author Response
The present scientific article is original and presents good quality in the structuring of the results. It has scientific merit and is of great importance to the literature. I strongly recommend the publication of the work after adapting it to the journal's norms.
A: We thank the reviewer for the positive response to this manuscript.